# Food Literacy and Dietary Intake in German Office Workers: A Longitudinal Intervention Study

**DOI:** 10.3390/ijerph192416534

**Published:** 2022-12-09

**Authors:** Svenja Meyn, Simon Blaschke, Filip Mess

**Affiliations:** Department of Sport and Health Sciences, Technical University of Munich, 80992 Munich, Germany

**Keywords:** health literacy, food literacy, dietary intake, nutrition, office workers, health promotion

## Abstract

Widespread patterns of poor dietary behavior are a key factor causing the increasing prevalence of chronic diseases around the world. Research has provided initial insights into the potential of food literacy (FL) to empower individuals to improve their dietary behavior. However, studies on FL interventions in working adults are scarce. The intervention delivered in this study was a comprehensive 3-week full time education-based workplace health promotion program (WHPP) that provided the participants with in-depth knowledge and skills regarding nutrition and health. We aimed to investigate the short- and long-term effects of the WHPP on FL and dietary intake (DI) and to examine the association between FL and DI in a sample of 144 German office workers (30.0% female). Using two random intercept mixed linear regression models, we found significant strong improvements for both FL (β = 0.52, *p* < 0.0001) and DI (β = 0.63, *p* < 0.0001) after the WHPP when compared to baseline. Significant long-term improvements at 18 months were strong for FL (β = 0.55, *p* < 0.0001) and weak for DI (β = 0.10, *p* < 0.0001). FL showed a significant moderate effect on DI across all measurement time points (β = 0.24, *p* < 0.0001). We conclude that well-designed WHPPs can induce long-term improvements in FL and DI, and that FL can be viewed as an asset to further expand food-related knowledge and skills and to enhance dietary behavior. Our study fills a gap of long-term findings regarding the role of FL in WHPPs and supports the idea of implementing FL in the development of comprehensive WHPPs to improve DI.

## 1. Introduction

The increasing prevalence of chronic diseases is challenging societies and healthcare systems around the world. Lifestyle habits such as dietary behavior play a key role in the development of various types of chronic diseases, including obesity, diabetes mellitus, cardiovascular disease, hypertension, stroke and some types of cancer [1]. In recent decades, Western populations have developed sociocultural habits and dietary patterns with detrimental health effects. Specifically, prevalent eating behavior has shifted from the consumption of whole foods to highly processed, energy-dense and nutrient-poor foods, also referred to as the nutrition transition [2,3,4]. Moreover, due to the abundance of convenience foods, many people no longer possess the necessary knowledge and skills to prepare healthy meals from whole food ingredients [3]. Together, these circumstances make it increasingly difficult for individuals to navigate the food environment in a healthy way [5]. Considering these extensive sociocultural nutrition-related threats to the health of individuals and societies, enabling people to make healthier food choices to prevent chronic disease is a key challenge in the modern world [1,6].

A large part of the population for which specific nutrition-related health threats have been investigated are office workers (OWs), who represent about 37% of all employees in Germany [7]. Specifically, pressure and stress at work, limited availability of healthy food and social influences encouraging the consumption of unhealthy foods have been found to act as barriers to healthy eating in office-based workplaces [8]. Similar research in various workplace settings confirms these findings and presents further barriers to healthy eating in the workplace, such as lack of self-control and convenience, work commitment, lack of time and cost of food [9,10,11,12]. Together, these barriers create a challenging environment for maintaining a healthy diet in the workplace.

In addition to these nutrition-related barriers to health-enhancing behaviors, OWs are highly exposed to sedentary behavior as a characteristic of their profession [13,14]. Together, nutrition and sedentary behavior result in an “energy gap”, a term coined in research to describe the interaction of increased energy intake and decreased energy expenditure, leading to increased rates of obesity and a range of non-communicable diseases [4,15,16].

Given this twofold burden of factors increasing their risk of chronic diseases, OWs need to be provided with knowledge and skills to improve their health behavior. Educating them on food and nutrition seems promising to accomplish this goal, since increased competencies can empower individuals to better navigate their food environment both inside and outside the workplace [17]. On the one hand, the resulting improved health behavior can be expected to increase employees’ health and well-being on the individual level [1]. On the other hand, these improvements can in turn be expected to also benefit a company through factors such as reduced sickness absences and increased work performance [18,19].

### 1.1. Introducing Food Literacy

Food literacy (FL) can be described as knowledge and skills related to nutrition [20]. The term has gained popularity among researchers in recent years, referring to a theoretical concept that can be addressed by interventions to affect health-related outcomes such as dietary behavior or, more specifically, dietary intake (DI) [4,21]. Early definitions of FL were based on Nutbeam’s model of health literacy, which includes functional, interactive and critical health literacy [22]. Functional FL refers to knowledge and skills regarding obtaining and understanding food-related information. Interactive FL comprises the ability to share such information in interactions between individuals, while critical FL focuses on the reflective analysis of the information obtained [23]. However, conceptualizations of FL have evolved over the years and researchers have not yet reached a consensus [24,25]. The present study is set in the context of chronic disease prevention in OWs and, thus, places the focus on the relations between FL and health-enhancing eating behavior in individuals. In accordance with the view of Krause et al. [26], we define FL as the “competencies needed to maintain a healthy diet” (p. 278).

### 1.2. Current State of Research on Food Literacy and Dietary Intake

People commonly assume that educating individuals on nutrition will help them improve their dietary behavior [27]. However, possessing food skills does not necessarily translate into healthy dietary behavior directly [28,29] because dietary behavior is also influenced by other factors such as stress [30], sleep duration [31] or the barriers to healthy eating in the workplace mentioned above [8]. Diverse influencing factors on food choice are also presented in a model by Chen and Antonelli [32]. This model includes food-internal factors (e.g., perceptual features), food-external factors (e.g., social environment), personal state factors (e.g., physiological needs) and cognitive factors (e.g., personal preference). The cognitive factors in this food choice model also include knowledge and skills, which overlap with FL. The model suggests that FL interacts with the other factors as well. Considering this variety of influences, it remains to be determined to what extent FL is actually associated with quality of DI with everyday challenges and whether targeted interventions can improve FL and DI.

Overall, the existing evidence on the effectiveness of FL interventions must be interpreted with caution. Firstly, a large amount of nutrition-related intervention research aimed at elements of FL does not use the term or concept. While these interventions do not discuss FL explicitly, their results and methodological discussions are still valuable for consideration in FL research due to the proximity of the underlying concepts. For example, reviews on the effectiveness of adult cooking interventions have concluded that diet-related knowledge and behavior can be improved [33,34,35]. However, there is agreement that the evidence is rather weak in quality and inconsistent regarding results, with quality issues including small sample sizes, deficient evaluation tools and lack of long-term follow-up data [33,34,35].

Secondly, many publications that refer to FL explicitly do not use a clear concept of the term, and publications that do use a clear concept have not reached a consensus regarding which concept to use [24]. Thus, readers should bear in mind that, on the one hand, there may be publications that do not use the term FL, but are still worth considering as background to FL research. On the other hand, there may be publications claiming to investigate FL that have little in common with the prevailing conceptualizations in FL research.

Regarding research on FL explicitly, two conceptual reviews from 2017 and 2019 [23,36] delineate a healthy diet as a consequence of FL, implying a general association between the two variables. Support for this assumption can be found in a systematic review by Vaitkeviciute and Harris [37], who investigated the relationship between FL and DI in adolescents. Promising results regarding FL interventions were reported by Begley et al. [27], who found that a four-week nutrition education and cooking program targeting low to middle-income Australians improved FL and dietary behavior. Moreover, a four-week online FL program targeting Australian adults improved FL and increased fruit and vegetable consumption [38]. However, a study by Rees et al. [39] on participants of a seven-week cooking program showed increased cooking confidence, but yielded no improvements in nutrition knowledge or healthy eating.

The controversy and limited evidence within this rather young field of research clearly reveal the need for further investigations on the effectiveness of interventions addressing FL to enhance dietary outcomes [26,36,40] and on the relation between FL and DI [6]. Given the health threats and barriers to healthy eating described above, this is especially the case for OWs [8].

When looking at interventions in the workplace, there are various reports of workplace health promotion and wellness programs. These programs include environmental modifications and education, or combinations of both. Focusing on the literature that includes education on dietary behavior, there are reviews and a meta-analysis evaluating worksite health promotion and wellness programs that have found evidence indicating that such programs can effectively improve DI, for example, by increasing fruit and vegetable intake and decreasing fat intake [41,42,43,44]. These reviews and the meta-analysis cover a wide range of work settings. One review also focuses on promoting healthy eating in OWs specifically; it supports the view that dietary interventions can be expected to yield positive effects [45]. However, the research mentioned focuses on dietary behavior, but not on FL. To our knowledge, no previous study has determined the effect of a workplace health promotion program (WHPP) on FL and DI and examined the relation between FL and DI in OWs.

### 1.3. Aims of the Research

To address the gap in previous research presented above, we investigated the 1.5-year long-term effectiveness of a 3-week full-time WHPP regarding FL and DI, as well as the relation between FL and DI in German OWs using four measurement time points. The corresponding research goals (RG) are:**RG1**: To determine whether a 3-week WHPP improved FL in German OWs and whether improvements remained stable over the course of 1.5 years;**RG2**: To determine whether a 3-week WHPP improved DI in German OWs and whether improvements remained stable over the course of 1.5 years;**RG3**: To investigate the extent to which FL acts as a predictor of DI.

## 2. Methods

### 2.1. Participants

The sample size was calculated a priori with a repeated measures within-between interaction analysis of variance design in G*Power 3.1 [46]. This calculation resulted in a sample size of *N*  =  98 participants to examine an effect of *f^2^* =  0.20 with a statistical power of β = 0.95 and a type one error of α  =  0.05 under the assumption of a repeated measures correlation of *r*  =  0.50 at four measurement times. Thus, *N* = 98 was the lower threshold for data collection, with an estimated attrition rate of about 50% [47].

The sample investigated in this study consisted of 144 OWs of a large engineering company in Germany. The sampling process consisted of three stages. The first stage was purposive sampling by offering participation in a 3-week full time WHPP to OWs for whom participation had been assessed as advisable based on a medical health check by a company physician. The second and third stage involved voluntary response sampling, where the OWs decided on participation in the WHPP and subsequently decided on participation in the study. Individuals with acute disease were not included in the study. Four hundred forty-six employees took part in the WHPP during the data collection period, of which 387 registered to participate in the study and 328 met the inclusion criteria of working in an office and not suffering from acute disease. Of these 328 individuals who filled in the questionnaire before the WHPP, 301 (92%) also answered it directly afterwards, 202 (62%) participated again after six months and 146 (45%) also completed the final measurement after 18 months. From this sample of 146 participants, we excluded one due to too many missing responses and removed another as a multivariate outlier, resulting in a final sample of 144 (44%) participants. Table 1 displays the sociodemographic parameters of the participants who completed all measurement time points, both for the complete sample and divided by gender.

### 2.2. Intervention

The 3-week full-time WHPP took place in a wellness hotel and was implemented with groups of 60 employees each, including either male or female participants only. The WHPP was instructed by physicians, physical education teachers, psychologists and a dietitian. The concept of the WHPP covered the facets of nutrition and physical activity in connection with health. In addition, the WHPP implemented the information–motivation–behavior skills (IMB) model by Fisher et al. [48], which assumes that information on the behavior, as well as the motivation and behavioral skills to act, are the determinants of health-related behavior and thus the keys to changing it. Table 2 displays the components of the WHPP with respect to the IMB model.

At the beginning of the WHPP, a medical health check by an independent physician served to determine the participants’ health status at the start of the WHPP and to define individual health goals in a participatory approach together with the physical education teachers and the dietitian. During the medical health check, motivational interviewing was employed to enhance the participants’ personal commitment to health behavior change.

During the following two weeks, the WHPP was structured with a daily morning activation, a 90-min exercise session after breakfast and a 90-min exercise session after lunch. With respect to nutrition, information on the nutrient content of the ingredients and their effects on the body were given with the menu. In addition, pictures presented next to the buffet showed ideal serving sizes and corresponding amounts of calories. The dietitian was available to provide additional information on meal preparation and cooking skills during lunch and dinner. Furthermore, the WHPP included presentations covering topics such as health risk and health behavior change strategies. The dietitian conducted a 4 h workshop on nutrition in the first week of the WHPP. This workshop covered a variety of food-related topics such as macronutrients, their role in the body, health-related effects, content in different foods and nutrition guidelines [49] and consumption amounts according to the recommendations of the German Nutrition Society [50]. In addition, individual coaching sessions with the dietitian, the physical education teachers and the psychologists were offered to the participants. These one-hour coaching sessions were scheduled in the second half of the WHPP with the aim to strengthen personal motivation for health behavior change goals and to develop behavioral skills for health behavior maintenance, for example, by identifying individual barriers to the desired health behaviors expected after the WHPP and developing corresponding strategies.

In the last week of the WHPP, another medical health check was conducted by the independent physician to examine objective changes in the participants’ health and to refine individual goals for the participants’ health behavior. In addition, the physician, the physical education teachers and the dietitian instructed the participants on problem-solving strategies in order to strengthen long-term motivation and behavioral skills to sustain changes in DI and physical activity in the personal environment of the OWs. Moreover, key information including presentation slides, nutrient tables and recipes was given to the participants after the WHPP for use at home. Further details on the WHPP can be found in the Appendix A and obtained from the corresponding author upon request.

### 2.3. Measurement Procedures

We conducted this research as part of a comprehensive evaluation of the occupational health management of a large engineering company in Germany. This engineering company has about 86,000 employees in Germany with more than 80% in office-based occupations. Data were collected between April 2019 and July 2021. For this study, we used data from questionnaires covering sociodemographic details, FL and DI. Assessments were conducted before the WHPP (T0), after completion of the WHPP (T1) and at 6- and 18-months follow-up (T2 and T3). The data at T0 and T1 were collected from April 2019 to December 2019 and follow-up data were collected from January 2020 to July 2021. Participants gave informed consent before filling in the questionnaires. In order to track individuals across measurement time points while ensuring data privacy, we asked them to generate a participant code that was based on four questions and designed to be easily reproducible by the participant. All study procedures complied with the German Federal Data Protection Act (BDSG), with the company’s data privacy guidelines and with the ethical standards of the 2013 Declaration of Helsinki [51]. The study received approval from the Ethics Committee of the School of Medicine of the Technical University of Munich (IRB number: 645/20 S-KH) and was pre-registered with the Open Science Framework (https://osf.io/wnts5, accessed on 4 December 2022).

### 2.4. Measures

The full questionnaire used in the study included details on sociodemographic characteristics, a questionnaire on FL and a questionnaire on DI. The full questionnaire is provided in the Appendix A.

(1)We evaluated FL using the Short Food Literacy Questionnaire (SFLQ) by Krause et al. [52]. As the SFLQ was originally validated with a Swiss sample, we adapted it slightly to fit German participants. The SFLQ is based on the theoretical framework of health literacy by Nutbeam [22] and assesses functional, interactive and critical FL. It consists of 17 items (e.g., “When I have questions on healthy nutrition, I know where I can find information on this issue.”, Cronbach’s α = 0.81, 95% CI = [0.78; 0.83]). The response format includes 4- and 5-point Likert scales and varies across the items. The responses for items 2a to 2e as well as for items 6a and 6b were each averaged before forming the overall score by summing these averaged values and the responses for the remaining items. The overall score ranges between 0 and 52, with higher scores indicating better FL [53].(2)Information on DI was collected using a German Food Frequency Questionnaire (FFQ) [54]. The FFQ enquires about consumption frequencies of 24 food groups (e.g., “How often do you eat cheese?”) with no indication of amounts. The response format is a 5-point Likert scale ranging from “1—nearly every day” to “6—never”, and the questions refer to the previous four weeks. For each food group, a score is assigned (0, 1, or 2) representing how closely the response matches the recommended consumption frequency according to the German Nutrition Society. The overall score is formed by summing the scores of 15 selected food groups (items 1, 2, 4–12, 14, 16, 17, 20). The score ranges between 0 and 30, with higher values indicating consumption frequencies closer to the recommendations. The reliability and validity were confirmed using a 7-day food diary in German adults [54].

### 2.5. Data Analysis

We conducted the statistical analysis in R (Version 4.2.0) [55]. The main data set included the participants who completed all four measurement time points and whose participant codes could be matched. This data set will be referred to as the per-protocol (PP) data set. We handled and reported missing data following the recommendations by Sterne et al. [56]. We imputed missing data using the mice package in R [57], applying predictive mean-matching under the assumption that data were missing at random. Of the sociodemographic variables, 10% were missing, having been neglected to be collected for those individuals. For the FFQ and the SFLQ, the proportions of missing item responses were 2.6% and 6.2%, respectively, with no known reason for the absence of responses. We input missing sociodemographic variables based on both SFLQ and FFQ item responses at T0. For FL and DI, we input missing SFLQ item responses based on all SFLQ items and missing FFQ item responses based on all FFQ items, each including all measurement time points. 

To examine whether missing values and drop-outs affected the results, we created a second data set containing the data of all individuals who completed at least T0 and T1, or T0, T2 and T3 with their participant codes assignable. This data set comprised a total sample of *N* = 303 and will be referred to as the intention-to-treat (ITT) data set. In the ITT data set, we input missing data as well, using predictive mean matching and assuming missingness at random. Sociodemographic variables were input from FFQ and SFLQ item responses at T0 using single imputation. Next, we added data from T1, T2 and T3 and calculated questionnaire sum scores for participants and measurement time points with no missing values at the item level. We input missing sum score values from existing sum score values and sociodemographic variables using multiple imputation with five data sets. This was the case for 17.1% of the SFLQ sum score values and for 18.0% of the FFQ sum score values. Trace plots and kernel densities indicated the good quality of the imputation. A descriptive comparison of the PP data set with one of the ITT data sets can be found in Table A1 in Appendix B.

Prior to the analysis, we removed multivariate outliers based on the recommendations by Tabachnick and Fidell [58]. That is, we calculated the Mahalanobis distance for each participant and used *p* < 0.001 as the cutoff to define multivariate outliers based on the resulting *X^2^*-distribution. This was the case for one individual. We analyzed all variables descriptively first, reporting mean and standard deviation (*M* (*SD*)) for all variables. Pairwise comparisons of means based on paired Wilcoxon signed-rank tests [59] and with *p*-values adjusted using the Bonferroni method [60] are displayed in Figure 1. The regression assumption of linearity was evaluated using a plot of residuals and fitted values, homoscedasticity was evaluated using a plot of residuals, normality was evaluated using a QQ-plot and independence of the residuals was evaluated using the Durbin–Watson test (Model 1: d = 2.0; Model 2: d = 2.1). All assumptions were met [61] and there was no considerable multicollinearity in the predictors according to the variance inflation factor (Models 1 and 2: VIF = 1.0). We set the significance level to α = 0.05 and reported unstandardized as well as standardized regression coefficients to allow for evaluating the effects in the development of FL and DI over the measurement time points. Effect sizes are classified as weak (β ≈ 0.10), moderate (β ≈ 0.30) or strong (β ≈ 0.50) [62], and each measurement time point is compared to T0. We evaluated the model fit following the recommendations by Nakagawa and Schielzeth [63], reporting the marginal *R^2^*, defined as the proportion of the variance explained by the fixed factors, and the conditional *R^2^*, defined as the proportion of the variance explained by the entire model including fixed and random effects.

We used two random intercept mixed linear regression models [64] to evaluate the research goals. Model 1 served to evaluate RG1 by investigating the development of FL over the four measurement time points. Model 2 served to evaluate RG2 by considering the development of DI over time while adjusting for FL, and to evaluate RG3 by determining the overall association of FL and DI while adjusting for the measurement time points. Specifically, the following variables were included in the models:**Model 1**: Measurement time point included as the predictor, FL as the outcome, age and gender as control variables, and the subject ID as a random factor.**Model 2**: FL included as the predictor, DI as the outcome, age, gender and measurement time point as control variables, and the subject ID as a random factor.

## 3. Results

Based on the descriptive analysis, both FL and DI scores increased when comparing post-WHPP scores to pre-WHPP scores. FL scores increased from 33.0 (5.9) at T0 to 38.6 (4.6) at T1, increased further to 39.4 (5.1) at T2 and remained at about the same level with 38.8 (5.2) at T3. DI scores increased from 13.7 (3.3) at T0 to 19.3 (2.7) at T1 and subsequently settled to 15.4 (3.4) at T2 and 15.3 (3.4) at T3. Trajectories of FL and DI scores over the different measurement time points are displayed in Figure 1, including significance levels for pairwise comparisons.

Model 1 yielded a marginal *R^2^* of 26.9% and a conditional *R^2^* of 63.5%. Regarding RG1, which aimed to determine the effect of the intervention on FL, we found a strong increase in FL at T1 (β = 0.52, 95% CI = [0.45; 0.60], *p* < 0.0001), as well as at T2 (β = 0.60, 95% CI = [0.53; 0.68], *p* < 0.0001) and T3 (β = 0.55, 95% CI = [0.47; 0.62], *p* < 0.0001). 

Model 2 yielded a marginal *R^2^* of 34.3% and a conditional *R^2^* of 64.0%. For RG2, which aimed to investigate the effect of the intervention on DI, we found a strong improvement in DI at T1 (β = 0.63, 95% CI = [0.55; 0.71], *p* < 0.0001) and a weak improvements at T2 (β = 0.10, 95% CI = [0.02; 0.18], *p* < 0.05) and T3 (β = 0.10, 95% CI = [0.02; 0.18], *p* < 0.05). With respect to RG3, which aimed to examine the capacity of FL as a predictor of DI, Model 2 showed a moderate overall effect of FL on DI (β = 0.24, 95% CI = [0.15; 0.33], *p* < 0.0001). 

Regression coefficients for the control variables and unstandardized regression coefficients for all variables are reported in Table A2 (Model 1) and Table A3 (Model 2) in Appendix B. Comparing the results from the PP data set to the results from the ITT data set did not yield considerable differences, supporting the assumption that missing data did not influence the results of the analysis. A comparison of the model results for the two data sets can be found in Table A2 (Model 1) and Table A3 (Model 2) in Appendix B. 

## 4. Discussion

Empowering people to make healthier food choices to prevent chronic disease is a key challenge in the modern world [1,6]. The goal of this study was to evaluate whether a 3-week WHPP improved FL and DI in German OWs, and whether FL was a determinant of DI. Overall, we found long-term improvements for both FL and DI and a moderate effect of FL on DI (Table A2 and Table A3). Results are discussed for each RG individually. 

### 4.1. Effect of the WHPP on Food Literacy

Our data show strong improvements in FL after the WHPP compared to before (Table A2). These findings align with previous literature showing that nutrition education and cooking programs can improve nutrition-related skills and knowledge in various populations [33,34,35]. While previous research focusing explicitly on FL has been promising regarding the effectiveness of interventions on FL in diverse adult populations [27,38,39], our findings strengthen the evidence by focusing on OWs as a population with an increased risk of chronic disease [8]. Establishing FL as a variable that can be improved through health promotion interventions is vital because research on the closely related and more extensively studied concept of health literacy states that health literacy can be viewed as an asset that empowers individuals to exert greater control over their health [65].

Interestingly, our results show even further increases in FL at T2. We assume that the underlying mechanism may be the higher proficiency in food-related knowledge and skills after the intervention, which in turn equips individuals with a base that makes accessing and understanding nutrition-related information easier. Support for this assumption can be found in research on health literacy. For example, a study found that individuals with higher levels of health literacy exhibit an increased use of the internet to seek health information compared with individuals with lower levels of health literacy [66]. Furthermore, the conceptualization of FL, which is based on the health literacy model by Nutbeam [22], holds that interactive FL comprises abilities to share nutrition-related information in interactions between individuals [23]. This concept suggests that FL enables individuals to improve their nutrition-related knowledge and skills further through social interactions. Thus, we assume that the FL gained from the intervention may have empowered our participants to subsequently further improve their FL, for example, by making use of other nutrition-related education offers within the company, by obtaining information from the internet, or by exchanging information with colleagues, friends and family [23,66].

However, the idea that higher levels of FL facilitate further improvements also comes with a flipside, because individuals with lower levels of FL tend to benefit less from interventions [65]. To attenuate this effect, interventions should be designed in accordance with the idea of precision prevention [67] by tailoring the content and methods to the competencies of the participants at baseline. This approach could be particularly valuable for WHPPs targeting diverse workforces, since social status and educational level have been found to correlate with health literacy [68]. For example, groups could be built according to baseline FL levels, or interventions could be designed to offer suitable individualized teaching content and methods for participants on all levels.

Overall, our findings suggest that interventions can improve FL if designed and conducted carefully and might empower participants to continue expanding their knowledge and skills after the intervention.

### 4.2. Effects of the WHPP on Dietary Intake

Our data show strong improvements in DI after the WHPP in comparison to the baseline level (Table A3), confirming the findings of previous studies evaluating nutrition-related effects of WHPPs [41,42,43,44]. Positive effects were still present in our study at both follow-up measurements. The significant long-term improvements are promising, since they indicate long-term stability of positive changes in the participants’ diets. These findings are valuable because it has been stated in previous literature on health behavior change that long-term effects of interventions are often either not reported [33,35], or in cases where follow-up measurements have been conducted, improvements were often not maintained by the participants [69,70]. 

When evaluating the trajectory of DI scores, each compared to T0, the strong increase at T1 in contrast with the small improvements at T2 and T3 is striking. This finding can most likely be attributed to the fact that the FFQ assesses DI retrospectively; thus, DI scores at T1 mainly refer to the intervention period. While the participants were free to choose their food at all times, the WHPP in the wellness hotel provided a food environment that strongly supported healthy choices.

In general, health behavior is influenced by personal and environmental factors [70,71,72]. Personal factors such as knowledge, skills and motivation in relation to eating a healthier diet or dealing with stress were addressed in the WHPP. However, the food environment which participants are exposed to in their everyday life was not been fundamentally changed by the program. Thus, a main reason for the difference in DI scores between T1 and the follow-up measurements can be assumed to lie in the effects of the different food environments, with the WHPP setting enhancing healthy dietary behavior, while the participants’ everyday setting presents a range of barriers to healthy eating [8,73].

In association with this, it should be pointed out that our study was focused on the empowerment of individuals to adopt a healthy lifestyle, thus primarily covering personal factors of health behavior. However, readers should be aware that comprehensive WHPPs should also address and modify environmental factors to avoid placing the entire burden on the individual and achieve maximized long-term improvements [18,70,71]. Healthy food needs to be made available, easily affordable and attractive in order to create health-enhancing food environments in which food-literate individuals can overcome remaining barriers and adopt healthy eating patterns [72].

All in all, our results on DI show that a WHPP focusing on personal factors of health behavior can be effective, but might be limited to small effects. Therefore, WHPPs focusing on personal factors should be combined with additional intervention components such as changes to the social network or the food environment [74]. Thus, again based on health literacy research [75], it would be advisable to not only focus on improving FL in employees, but also on establishing guidelines for food literate companies. 

### 4.3. Associations of Food Literacy with Dietary Intake

Our data show a significant moderate effect of FL on DI across all measurement time points (Table A3). This finding suggests that knowledge and skills related to healthy eating do influence dietary behavior and provides support for delineating DI as a consequence of FL, as proposed by previous conceptual reviews [23,36]. A systematic review on the relationship between FL and DI in adolescents has found evidence for such a relationship as well, but the included studies have been reported to be limited in their validity due to mostly cross-sectional designs and the lack of validated measurement tools [37]. To our knowledge, similar findings in adult OWs have not been established yet. Our study contributes to strengthening the evidence on the relationship between FL and DI in adults.

Moreover, our study adds to the literature on workplace health promotion and wellness programs, which so far largely focuses on measuring dietary behavior as an outcome. By additionally investigating FL, we provide preliminary evidence for a potential mechanism explaining the effect of education-based WHPPs on DI. This mechanism strengthens the conceptualization of FL as an asset [65] to support the adoption of a healthier diet. That is, FL could help OWs overcome barriers to healthy eating in the workplace [8] and to attenuate the energy gap [76] that results from sedentary work in combination with increased energy intake in order to achieve improved health outcomes [1]. We hope that our research will facilitate the inclusion of measures that address FL in the design of evidence-based health promotion interventions for adult OWs and encourage funders to invest in such interventions [77]. 

However, the effect of FL on DI that we found is moderate, suggesting that DI is affected by other factors as well. In fact, researchers propose an indirect relationship between FL and nutrition, which is influenced by factors such as individual values and food supply, as well as sociocultural factors and environmental aspects such as availability and affordability [17,29,72]. Motivation, which has not been measured in this study, is also considered essential for health-enhancing behavior in the IMB model by Fisher et al. [48]. Thus, FL can be regarded as one personal factor in a variety of personal and environmental variables that influence DI.

We conclude that interventions should not solely address FL but rather a combination of the influencing variables in order to achieve the most beneficial improvements in health-enhancing dietary behavior. That is, interventions should not only focus on teaching knowledge and skills but also incorporate key aspects such as promoting motivation, ideally based on psychological behavior change models such as the IMB model [78].

### 4.4. Strengths and Limitations

We conducted this analysis using data from a comprehensive evaluation of a WHPP for employees of a large German engineering company. Strengths of our study include four measurement time points, including 6- and 18-month follow-ups, allowing the evaluation of the long-term effectiveness of the WHPP, which has been pointed out to be lacking in other diet-related programs [33]. The use of both a PP analysis and an ITT analysis based on multiple imputation further increases the quality of our results [56].

Regarding measurement tools, the use of the SFLQ and the FFQ as validated questionnaires to measure FL and DI can be considered a strength [34]. Moreover, the SFLQ is based on a widely accepted theoretical framework of health literacy by Nutbeam et al. [22] and was developed following a methodologically sound scientific procedure [52]. 

The 3-week full-time WHPP was a professionally designed intervention that seems well-suited for research on FL due to its high intensity, fairly long duration and comprehensive concept covering all levels of the IMB model of health-related behavior change by Fisher et al. [48]. Moreover, with the intervention taking place in a wellness hotel, it benefitted from a positive implementation climate and the high engagement of the participants [79].

On the other hand, the design of our study also comes with some limitations. Since the data were collected in the context of the WHPP, there were no resources available to include a control or comparison group in the study design. Thus, we cannot evaluate whether our intervention was more effective compared to, for example, a shorter intervention using fewer resources. Additionally, the limitations of self-rating questionnaires need to be considered when interpreting our results, since respondents may not always have estimated and reported their knowledge, skills and behaviors accurately [28,53]. Specifically, distortions such as response-shift bias [80], recall bias, confirmation bias and social desirability bias [81] may have influenced the results.

In addition, our results may have been influenced by the COVID-19 pandemic. While the intervention as well as the T0 and T1 measurements were conducted before the pandemic, the measurements at T2 and T3 may have been influenced by general shifts in eating behavior due to the extensive restrictions on everyday life that were implemented starting in early 2020 [82]. For example, a systematic review of longitudinal studies on eating behavior changes during the COVID-19 pandemic concluded that adherence to healthy diets decreased during that time [83]. This effect may have attenuated the positive long-term effects of the WHPP on DI.

Readers should also be aware that the findings of this study are based on a sample of participants who were working in a large German engineering company, of which 70% were male, most were aged around 50 and most had a higher educational background. Thus, inferences to other populations should be considered carefully.

Moreover, it should be clarified that in the context of the existing literature on FL, we apply a rather narrow definition of FL in order to match our study goals of investigating FL and DI in the context of chronic disease prevention in OWs. While this definition focuses on the individual, several researchers suggest extending the conceptualization. They argue that food-related attitudes, knowledge and skills not only affect the person but also interact with the entire food system including social, cultural, economic and environmental issues, and that these aspects should therefore be considered in definitions of FL [84,85,86,87]. A range of concepts of FL are being applied in the literature; thus, readers should be aware of which concept each publication is based on [20].

### 4.5. Suggestions for Future Research

Firstly, considering the limitations of self-rated questionnaires, future studies could investigate whether the use of objective measures and the inclusion of a control group will yield results similar to our findings. Provided sufficient funds and resources are available, FL could be measured using task-based items [28] and DI could be measured using food records over several days, against which FFQs are frequently validated [88]. Moreover, the inclusion of comparison groups could strengthen the scientific evidence and provide insights into the respective effectiveness and efficacy of interventions that differ in characteristics such as duration, intensity and level of customization within the framework of precision prevention.

A finding worth investigating in future studies is that the participants’ FL increased even further after the end of the study. Randomized controlled trials should be conducted to verify our findings. Such trials could also provide further insights into whether the increased FL induced by the intervention is indeed the cause for further improvements in FL after the intervention or to what extent other mechanisms, such as enhanced awareness of the effects of nutrition on individual health or increased motivation, play a role. Findings on these underlying mechanisms that empower individuals to continue improving their FL after an intervention could then provide information for the design of future FL interventions.

Regarding DI, our study focused on the basic relationship between FL and DI. Building upon these findings, it would also be interesting to investigate the relation of FL and DI with other factors that influence DI using structural equation modeling. For example, the potential moderating effects of motivation [48], stress [30], sleep [31] and barriers to healthy eating in the workplace [8] on the relationship between FL and DI could be examined.

## 5. Conclusions

Our study indicates that health promotion programs can be effective with regard to FL and DI. We suggest that FL can be viewed as an asset that, on the one hand, equips individuals with a base that facilitates accessing, understanding, sharing and further expanding food-related knowledge and skills. On the other hand, FL can also empower individuals to improve their dietary behavior, thus acting as a particularly valuable resource to support OWs in overcoming barriers to healthy eating in the workplace and attenuating the energy gap to improve a range of health outcomes. Considering our findings and the associated literature, we suggest that effective health promotion interventions should ideally be based on psychological behavior change models and cover both personal and environmental factors that influence dietary behavior according to scientific evidence in order to achieve beneficial health outcomes.

## Figures and Tables

**Figure 1 ijerph-19-16534-f001:**
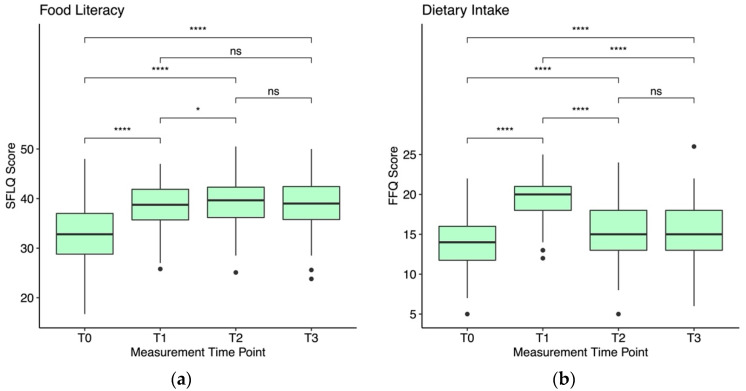
(**a**) shows standard box blots for FL scores measured by the SFLQ for each measurement time point. (**b**) shows standard box plots for DI scores measured by the FFQ for each measurement time point. Significance levels are denoted as ns: *p* > 0.05, *: *p* ≤ 0.05, ****: *p* ≤ 0.0001.

**Table 1 ijerph-19-16534-t001:** Participants’ sociodemographic characteristics.

	Total (*N* = 144)	Men (*n* = 101)	Women (*n* = 43)
Age, *M (SD)*	50.0 (6.3)	50.5 (6.2)	48.6 (6.6)
Relationship status			
Relationship, *n* (%)	106 (73.6)	77 (76.2)	29 (67.4)
Single, *n* (%)	38 (26.4)	24 (23.8)	14 (32.6)
Education			
Tertiary, *n* (%)	87 (60.4)	75 (74.3)	12 (27.9)
Secondary, *n* (%)	54 (37.5)	26 (25.7)	28 (65.1)
Primary, *n* (%)	3 (2.1)	0 (0.0)	3 (7.0)

Notes. *N* = total sample; *n* = sub-sample; *M* = mean; *SD* = standard deviation; tertiary = college degree; secondary = vocational training; primary = high school qualification.

**Table 2 ijerph-19-16534-t002:** Components of the WHPP with respect to the information–motivation–behavioral skills (IMB) model by Fisher et al. [48].

WHPP Component	Information	Motivation	Behavior Skills
-Medical health checks including motivational interviewing	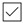	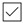	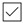
-Meals: nutrient content, serving sizes, consultation with dietitian	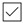		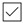
-Presentations and workshop	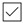	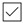	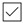
-Individual coaching sessions	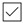	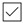	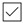
-Take-home information	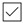		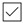

Notes. Checked cells indicate that this component of the WHPP covers the corresponding determinants of the IMB model.

## Data Availability

The datasets generated and analyzed during the study are not publicly available owing to patient confidentiality, but are available in a highly anonymized form from the corresponding author on reasonable request.

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
