# Peer review of "Food Literacy and Dietary Intake in German Office Workers: A Longitudinal Intervention Study"

_ijerph, 2022, doi:10.3390/ijerph192416534_

Round 1

Reviewer 1 Report

Review report

Title: Food Literacy and Dietary Intake in German Office Workers: 2 A Longitudinal Intervention Study

Journal : The International Journal of Environmental Research and Public Health

The paper focused on very important and interesting aspects of Food Literacy and Dietary Intake for the workers. The paper is well written and comprehensive, but the authors should address few points as follows:

1)                  In line “310” please add the reference of your findings (e.g., in Table 2 etc.).

2)                  In line “308” modify it.

3)                  Carefully check for typos, spacing issues, grammar usage and English language in whole manuscript.  

4)                  Please add Questionnaire for that study.

5)                  Please replace arm ratio with Arm Ratio.

6)                  In Lines “21”Please add conclusion in the abstract.

Author Response

Response to Reviewer 1

Dear reviewer,

Thank you very much for your feedback! We appreciate the valuable comments and the precious time to review our manuscript. It was your valuable and insightful comments that led to possible improvements in the current version. We have carefully considered the comments and tried our best to address and implement every one of them.

The paper focused on very important and interesting aspects of Food Literacy and Dietary Intake for the workers. The paper is well written and comprehensive, but the authors should address few points as follows:

  • In line “310” please add the reference of your findings (e.g., in Table 2 etc.).

Response to 1): We are grateful for this comment and have added references in that line and adapted the other paragraphs of the discussion in the same manner.

  • In line “308” modify it.

Response to 2): Thank you for this note. However, we are not exactly sure if this is what you mean, but we added reference to the appendix tables in this line as well and reworded the line.

  • Carefully check for typos, spacing issues, grammar usage and English language in whole manuscript.

Response to 3): We truly appreciate this comment and have checked the manuscript precisely for typos, spacing issues, grammar usage and English language during the revision procedure.

  • Please add Questionnaire for that study.

Response to 4): Thanks, for this remark! We have submitted the questionnaire as supplementary material.

  • Please replace arm ratio with Arm Ratio.

Response to 5): Thank you very much for this comment, but the term “arm ratio” does not appear in our manuscript. Therefore, we did not address this comment.

  • In Lines “21”Please add conclusion in the abstract.

Response to 6): We appreciate this comment and have added a conclusion to the abstract.

Reviewer 2 Report

Dear Authors,

Thank you for giving me the opportunity to review your manuscript. The manuscript can be considered for publication following the clarification of the following points:

1.       Abstract: Abstract should conceptually focus on what was done, what was found and what was concluded. Clarification is needed about what was done (e.g. What was WHPP delivered). Furthermore, clarification is needed on what was concluded (e.g. the study fills a gap, in what way? What is the conclusion of this very study). Please revise the abstract to strengthen the above.

2.       Line 64, the definition of food literacy is simplistic and muddles. Please add a table with different definitions provided within the literature with reference, and clarify the rationale for selecting the definition used in this study. For instance, please see PMID: 28487244, PMID: 29914503, and PMID: 26280794, while there are more references available on this.

3.       Section 1.2, Lines 79-85, please provide a clearer outline of the role of FL within the Food Choice models. For this, I strongly encourage you to use/produce a schematic to clarify the contribution of FL to Food Choice.

4.       Section 1.2 There must be distinct paragraph/s about the measurement and scoring FL, as this is currently muddled while embedded through the text.

5.       Section 1.1 and 1.2 do not provide summary of the literature about the nutritional health promotion interventions previously delivered to office workers. If the body of knowledge is scarce (a simple search in Pubmed highlights a few article straight away…), this can more broadly consider health promotion interventions for office workers. Should word count be of concern, this can be produced within a table with Author, Year, Main Findings, etc.

6.       Section 2.1, clarify sampling method, sample size calculation, sample’s pool to establish the external validity.

7.       Lines 158-164, Please provide a schematic to clarify how intervention fit the IMB model by Fisher et al.

8.       Lines 165-182, please provide figures/sample within the article or supplementary material to clarify the WHPP delivered.

9.       Line 188, please clarify the potential impact of global pandemic during this time.

10.   Section 2.5 please clarify how assumption of normal distribution and homogeneity of the data was tested.

11.   Discussions, please provide reflection on how WHPP can be improved.

Author Response

Response to Reviewer 2

Dear reviewer,

Thank you very much for your feedback! We appreciate the valuable comments and the precious time to review our manuscript. It was your valuable and insightful comments that led to possible improvements in the current version. We have carefully considered the comments and tried our best to address and implement every one of them.

Thank you for giving me the opportunity to review your manuscript. The manuscript can be considered for publication following the clarification of the following points:

  • Abstract: Abstract should conceptually focus on what was done, what was found and what was concluded. Clarification is needed about what was done (e.g. What was WHPP delivered). Furthermore, clarification is needed on what was concluded (e.g. the study fills a gap, in what way? What is the conclusion of this very study). Please revise the abstract to strengthen the above.

Response to 1): Thank you for this helpful comment. We have revised the abstract and clearly outlined what was done and what was concluded.

  • Line 64, the definition of food literacy is simplistic and muddles. Please add a table with different definitions provided within the literature with reference, and clarify the rationale for selecting the definition used in this study. For instance, please see PMID: 28487244, PMID: 29914503, and PMID: 26280794, while there are more references available on this.

Response to 2): This comment is truly appreciated and understand the importance of clarifying the rationale for selecting a specific definition. We have intentionally chosen this particular definition, since we consider it to be the one that fits best with our study because the study is focused on promoting healthy eating in individuals and has a focus on the personal perspective of health promotion. We have also provided this rationale for choosing our definition in section “1.1. Introducing Food Literacy”. The articles that you are suggesting are all referenced in the manuscript, and additionally we have discussed the definition in the limitations section of the paper. Lists of definitions are provided elsewhere in the literature that we are referencing (e.g., doi.org/10.3390/nu13062006), thus we do not think it is necessary or expedient to provide another one in our manuscript. We hope you can understand our reasoning on this.

  • Section 1.2, Lines 79-85, please provide a clearer outline of the role of FL within the Food Choice models. For this, I strongly encourage you to use/produce a schematic to clarify the contribution of FL to Food Choice.

Response to 3): Thank you for this suggestion. We added a section addressing the connection of the food choice model and FL and put FL in context within the model. A schematic of the model is provided in the publication that we referenced.

  • Section 1.2 There must be distinct paragraph/s about the measurement and scoring FL, as this is currently muddled while embedded through the text.

Response to 4): We are grateful for this comment and have now created distinct paragraphs for the FL and DI questionnaires, and added some additional information on the scoring as well as providing the questionnaire used in the supplementary files.

  • Section 1.1 and 1.2 do not provide summary of the literature about the nutritional health promotion interventions previously delivered to office workers. If the body of knowledge is scarce (a simple search in Pubmed highlights a few article straight away…), this can more broadly consider health promotion interventions for office workers. Should word count be of concern, this can be produced within a table with Author, Year, Main Findings, etc.

Response to 5): Thanks a lot for the comment. We extended the literature based on your suggestions and have referenced one more review that is focused on office workers specifically. In general, we have focused on referencing reviews, since we consider this to be most suitable for efficiently describing the essence of previous literature. In addition, we have also clarified in the manuscript that we focused on the literature that includes at least in part facets of food literacy or education on dietary behavior, since some of the articles focus on modifications of the environment and are therefore not in depth included in our introduction.

  • Section 2.1, clarify sampling method, sample size calculation, sample’s pool to establish the external validity.

Response to 6): This comment is largely appreciated, we added the sampling method, sample size calculation and described the overall sampling pool of the company

  • Lines 158-164, Please provide a schematic to clarify how intervention fit the IMB model by Fisher et al.

Response to 7): Thank you very much for this comment! We added a schematic on the implementation of the IMB model by Fisher et al. in the intervention. In addition, we described the intervention in greater detail to simplify the understanding of this implementation.

  • Lines 165-182, please provide figures/sample within the article or supplementary material to clarify the WHPP delivered.

Response to 8): We appreciate this remark. We added an overview of the WHPP in the supplementary file and a more detailed description of the WHPP

  • Line 188, please clarify the potential impact of global pandemic during this time.

Response to 9): Thank you, that is indeed a good point. Since participant recruitment and the intervention itself were completed before the pandemic, we found it most fitting to discuss potential effects of the pandemic on the follow-up (T2 and T3) measurements and consequently on the results in the limitations of our manuscript.

  • Section 2.5 please clarify how assumption of normal distribution and homogeneity of the data was tested.

Response to 10): Thanks for this comment! We have added the methods for all assumption checks.

  • Discussions, please provide reflection on how WHPP can be improved.

Response to 11): Thank you very much for this comment. We have included multiple aspects regarding the improvement of WHPPs throughout the discussion, including precision prevention, the modification of environmental factors, basing interventions on scientific behavior change models, as well as including long-term follow up measurements in the evaluation, and using objective measurement tools if feasible. We also highlight this again in our conclusion. Additionally, we have extended the reflection at the end of section 4.2. In our view, we have thus included a range of valuable aspects.

Reviewer 3 Report

Dear Authors,

Thank you for your manuscript. The study is well-designed, and the paper is very well-written.

The Introduction section provides comprehensive evidence from the literature on the existing knowledge of food literacy and obstacles to healthy eating among office workers. The section ends with three study aims.

Nevertheless, the authors themselves highlight the methodological study limitations: non-controlled intervention study design and food frequency questionnaire used, which is not sensitive enough to dietary habits changes. Moreover, the sample is specific and non-representative. The sample size calculation is not provided. 

I would suggest adding ethical procedures to the Methods section: information on ethics committee approval and procedures for obtaining informed consent from study participants.

Author Response

Response to Reviewer 3

Dear reviewer,

Thank you very much for your feedback! We appreciate the valuable comments and the precious time to review our manuscript. It was your valuable and insightful comments that led to possible improvements in the current version. We have carefully considered the comments and tried our best to address and implement every one of them.

Thank you for your manuscript. The study is well-designed, and the paper is very well-written.

The Introduction section provides comprehensive evidence from the literature on the existing knowledge of food literacy and obstacles to healthy eating among office workers. The section ends with three study aims.

  • Nevertheless, the authors themselves highlight the methodological study limitations: non-controlled intervention study design and food frequency questionnaire used, which is not sensitive enough to dietary habits changes. Moreover, the sample is specific and non-representative.

Response to 1): We appreciate this comment and discussed in detail the limitations with respect to the non-controlled intervention study design, the food frequency questionnaire employed and non-representative nature of the sample in the manuscript. Please let us know, if there are aspects, which you would like us to elaborate in more detail.

  • The sample size calculation is not provided.

Response to 2): Thank you very much for this comment! We have added the sample size calculation to the manuscript.

  • I would suggest adding ethical procedures to the Methods section: information on ethics committee approval and procedures for obtaining informed consent from study participants.

Response to 3): we are grateful for this remark. Information on ethics committee approval and procedures for obtaining informed consent from study participants is provided in section 2.3. Measurement Procedures.

Reviewer 4 Report

The work is a longitudinal study on improving food literacy and thus food intake.

We know that there are few studies on this topic. So the study is promising. I found the study well designed and the paper well written. I have some concerns and suggestions that could be taken up by the authors in a review. I hope you find them helpful.

ABSTRACT - I think the conclusion in the abstract could be more practical. Is it worth improving workers' FL? 

INTRODUCTION - The authors have done a good job in the introduction. Very clear!

L129 - Perhaps the research project should be changed to research. The word project seems to be a work at an early stage.

METHOD - The sample is from a single company/firm? Please elaborate 

I am confused about WHPP. They were in a hotel for 3 weeks and heard about health 8 hours a day?

Please indicate how you reached the IMB model. Based on this text, the intervention seems very superficial and based only on lectures. I think this section needs to be reworded 

Are there no guidelines for nutrition in Germany? Was this not used? 

RESULT - I found the figures a bit hard to read. Do not you think it would be better to use letters to compare the groups? 

Please indicate which posthoc test you used in Figure 1. If you used the Bonferroni correction, why did you find some groups significant with P < 0.05? Based on the Bonferroni correction (0.05/6 = 0.008), only p < 0.008 would be significant. Please clarify.

I strongly suggest to include descriptive results of the SFLQ and the FFQ. The authors were too direct with the regression. Which items of the SFLQ had higher and lower scores? For which foods did intake increase or decrease during the period?

DISCUSSION - The discussion is very good.

Author Response

Response to Reviewer 4

Dear reviewer,

Thank you very much for your feedback! We appreciate the valuable comments and the precious time to review our manuscript. It was your valuable and insightful comments that led to possible improvements in the current version. We have carefully considered the comments and tried our best to address and implement every one of them.

The work is a longitudinal study on improving food literacy and thus food intake.

We know that there are few studies on this topic. So the study is promising. I found the study well designed and the paper well written. I have some concerns and suggestions that could be taken up by the authors in a review. I hope you find them helpful.

  • ABSTRACT - I think the conclusion in the abstract could be more practical. Is it worth improving workers' FL?

Response to 1): This comment is truly appreciated, thank you! We revised the abstract based on your suggestion and included a statement on the value of FL for WHPPs.

INTRODUCTION - The authors have done a good job in the introduction. Very clear!

  • L129 - Perhaps the research project should be changed to research. The word project seems to be a work at an early stage.

Response to 2): Thanks for this remark. We changed the phrase “research project” to “research”

  • METHOD - The sample is from a single company/firm? Please elaborate

Response to 3): We are grateful for this comment. Owing to your question we presented this information in the section “Measurement Procedure”, but included this information due to its relevance also in the “Participants” section.

  • I am confused about WHPP. They were in a hotel for 3 weeks and heard about health 8 hours a day?

Response to 4): Thank you very much for this question. We described the WHPP in more detail and added relevant additional information on the intervention components and the respective purpose for the 3-week WHPP.

  • Please indicate how you reached the IMB model. Based on this text, the intervention seems very superficial and based only on lectures. I think this section needs to be reworded.

Response to 5): This comment is very much appreciated. We have added more detail to the description of the intervention, as well as a table indicating components of the WHPP and corresponding elements of the IMB model.

  • Are there no guidelines for nutrition in Germany? Was this not used?

Response to 6): Thank you very much! Guidelines and consumption recommendations of the German Nutrition Society were used in the WHPP. We changed the respective sections and added this information with references.

  • RESULT - I found the figures a bit hard to read. Do not you think it would be better to use letters to compare the groups?

Response to 7): Thank you very much for the comment. However, there were no groups in our study because we could only employ a single arm longitudinal design.

  • Please indicate which posthoc test you used in Figure 1. If you used the Bonferroni correction, why did you find some groups significant with P < 0.05? Based on the Bonferroni correction (0.05/6 = 0.008), only p < 0.008 would be significant. Please clarify.

Response to 8): We appreciate this question and understand the suggestion behind this comment. We have described the procedure in the methods section “2.5. Data Analysis”. We chose to adjust the p-values according to the Bonferroni method instead of adjusting the confidence level, which gives the exact same result and is intended to avoid confusing readers with uncommon significance levels in the figure caption.

  • I strongly suggest to include descriptive results of the SFLQ and the FFQ. The authors were too direct with the regression. Which items of the SFLQ had higher and lower scores? For which foods did intake increase or decrease during the period?

Response to 9): Thank you for this comment! We provided a table of the descriptive item results in the supplementary material.

DISCUSSION - The discussion is very good.

Round 2

Reviewer 4 Report

The study is improved. I think it can be accepted in present form.